# Crosstalk between Gut Microbiota and Host Immunity: Impact on Inflammation and Immunotherapy

**DOI:** 10.3390/biomedicines11020294

**Published:** 2023-01-20

**Authors:** Connor Campbell, Mrunmayee R. Kandalgaonkar, Rachel M. Golonka, Beng San Yeoh, Matam Vijay-Kumar, Piu Saha

**Affiliations:** 1Department of Physiology & Pharmacology, University of Toledo College of Medicine, Toledo, OH 43614, USA; 2Department of Physiology & Pharmacology, University of Toledo College of Medicine and Life Sciences, Toledo, OH 43614, USA

**Keywords:** gut microbiota dysbiosis, innate immune system, adaptive immune system, infection, cancer, inflammatory bowel diseases, fecal microbiota transplantation

## Abstract

Gut microbes and their metabolites are actively involved in the development and regulation of host immunity, which can influence disease susceptibility. Herein, we review the most recent research advancements in the gut microbiota–immune axis. We discuss in detail how the gut microbiota is a tipping point for neonatal immune development as indicated by newly uncovered phenomenon, such as maternal imprinting, in utero intestinal metabolome, and weaning reaction. We describe how the gut microbiota shapes both innate and adaptive immunity with emphasis on the metabolites short-chain fatty acids and secondary bile acids. We also comprehensively delineate how disruption in the microbiota–immune axis results in immune-mediated diseases, such as gastrointestinal infections, inflammatory bowel diseases, cardiometabolic disorders (e.g., cardiovascular diseases, diabetes, and hypertension), autoimmunity (e.g., rheumatoid arthritis), hypersensitivity (e.g., asthma and allergies), psychological disorders (e.g., anxiety), and cancer (e.g., colorectal and hepatic). We further encompass the role of fecal microbiota transplantation, probiotics, prebiotics, and dietary polyphenols in reshaping the gut microbiota and their therapeutic potential. Continuing, we examine how the gut microbiota modulates immune therapies, including immune checkpoint inhibitors, JAK inhibitors, and anti-TNF therapies. We lastly mention the current challenges in metagenomics, germ-free models, and microbiota recapitulation to a achieve fundamental understanding for how gut microbiota regulates immunity. Altogether, this review proposes improving immunotherapy efficacy from the perspective of microbiome-targeted interventions.

## 1. Introduction

‘No man is an island’, said John Donne, to describe relations between a human being and society [1]. However, this is also true when describing human metabolism. From birth, humans, like all other animals, are colonized by microbes, especially on the skin and mucosal surfaces, such as the gastrointestinal tract (GIT). The GIT harbors a substantial collection of microorganisms known as the gut microbiota. It is a balanced composition of over 5000 species encompassed under bacteria (e.g., 99% of the intestinal microbiota is composed of Firmicutes, Bacteroidetes, Proteobacteria, and Actinobacteria), fungi (e.g., Candida), viruses (e.g., bacteriophages), and parasites (e.g., flagellates) [2,3,4,5,6,7,8]. The gut microbiota acts like a ‘superorganism’ inside the human host and aids in the assimilation of food, produces metabolites that nourish the host, protects the host from infection, maintains function and morphology of intestinal epithelial cells, and regulates host immunity [4,8,9,10,11,12]. Under healthy conditions, the gut microbiota is in a balanced state of ‘eubiosis’. However, during diseased conditions, the gut microbiota enters an imbalanced state of dysbiosis in which there is either a bloom of opportunistic pathogens or a reduction in beneficial commensals, or both. The beauty of the host–microbiota relationship lies in the fact that microbes shape every aspect of human metabolism. As such, in addition to digestive and skin disorders, the gut microbiota has the potential to influence the pathogenesis of diseases, such as obesity and asthma, and psychological disorders, such as Parkinson’s disease [13,14].

Gut microbiota crosstalk with host immunity is one of the major features for physiological stability and a mechanism for disease etiology. There are two branches of the immune system, i.e., innate and adaptive, that work together to protect the body from external and internal threats. The innate immune system is the ‘first line of defense’ and provides fast non-specific responses upon an immunological stimulus. Innate immunity involves granulocytes, natural killer cells, dendritic cells, and macrophages that engulf the pathogen and secrete cytokines and chemokines. In addition to recruitment of more innate immune cells, cytokines attract lymphocytes, i.e., B cells, which produce antibodies unique to the specific pathogenic insult, and T cells (generally categorized into helper T cells, cytotoxic T cells, and regulatory T cells (Treg cells)), both of which form the basis of adaptive immunity [15,16]. Both arms of the immune system are tightly regulated to avoid extremes of over-activation or exhaustion, for which the gut microbiota is an essential factor (summarized in Graphical Abstract). In this review, we provide an in-depth outline and discussion about how the gut microbiota as a whole, in addition to specific bacterial species and microbial-derived metabolites, regulates immune responses. We further discuss how the gut microbiota–immune axis is aberrant in prevalent chronic inflammatory diseases and how modulation of the gut microbiota could be a therapy or possible adjuvant for other current treatments.

## 2. Role of Gut Microbiota and Their Metabolites in Neonatal Immune System Development

The first microbial colonization in a neonate depends on the mode of delivery (C-section vs. vaginal delivery) and feeding (formula vs. maternal milk) [17,18,19]. For instance, formula feeding was found to lower the diversity of the gut microbiota and expand pathogenic bacteria, such as *Enterobacteriaceae* and *Enterococcaceae*; this gut microbiota dysbiosis contributed to greater mucosa inflammatory activity and worsened pathology in a necrotizing enterocolitis model [20]. Moreover, a possibility for C-section to disrupt mother-to-neonate transmission of specific microbial strains (e.g., LPS-expressing bacteria) was reported [21]. However, the neonatal immune system may be primed during intra-uterine development since microbes generally present in maternal gut and mouth, such as Firmicutes, Actinobacteria, and Proteobacteria, are found in the placenta, umbilical cord, and amniotic fluid [22,23]. While an in utero microbiome is still under investigation, a 2020 article by Rackaityte et al. proposes that bacterial colonization would be limited in the human intestine in utero [24]. Moreover, recent evidence for an in utero intestinal metabolome was delineated and found to be enriched with amino acids (e.g., tryptophan), vitamins (e.g., riboflavin), and, more interestingly, gut-microbiota-derived bile acids [25].

The hygiene hypothesis proposes that exposure to a plethora of microbes early in life is essential to develop a robust immune system [26]. During intra-uterine development, the fetal innate immune system is suppressed by Foxp3^+^ CD4^+^ Treg cells to prevent immune development against maternal antigens [12]. At and after birth, antigens from commensals are recognized by several pattern recognition receptors (PRRs), such as Toll-like receptors (TLRs), on intestinal epithelia, resulting in less production of antimicrobial peptides and establishment of immune tolerance [27]. Alongside these, Paneth cells produce antimicrobial peptides, such as phospholipase-2, lysozyme, and defensins, but these molecules do not act against commensals and rather protect the neonatal gut from opportunistic pathogens [22,28]. *Bifidobacteria* spp. is one of the major commensals that impact infantile immunity, such as T cell maturation [29]. The absence of *Bifidobacteria* resulted in the depletion of human milk oligosaccharide production and was associated with greater Th2/Th17 immune activation [30]. It is noteworthy that formula feeding is associated with less *Bifidobacteria* abundance, but the effect is transient [31]. After lactation, pups undergo a newly defined process called ‘weaning reaction’, which is a shift in the gut microbiota that occurs when the offspring transitions from breast milk to solid food [32]. Weaning reaction was found to increase bacterial and dietary metabolites, such as short-chain fatty acids (SCFA) and retinoic acid [32]. Inhibition of weaning causes a pathological imprinting for increased risk to allergic inflammation and colitis [32]. This matches other reports that the absence of early exposure to microbiota can induce immunoglobin E (IgE) over-production and hypersensitivity to a wide array of antigens, which leads to conditions such as asthma and inflammatory bowel diseases [33,34,35]. Overall, early immune system development is regulated by the gut microbiota and can have a long-lasting impact on disease susceptibility.

## 3. Interaction between Gut Microbiota and Host Innate Immune System

The interaction between the gut microbiota and host mucosal immune system is critical in maintaining host health because it is the first line of defense against encroaching gut microbes (summarized in Graphical Abstract). The mucosal surfaces are compartmentalized with immune responses, including a dense mucus layer, tight junction proteins, and antimicrobial proteins. Intestinal innate immune cells develop tolerance to commensal bacteria by identifying invasive pathogens and preventing their passage from the intestinal lumen to circulation [36]. After trespassing through the epithelial barrier, invasive bacteria and pathogen-associated molecular patterns (PAMPs, i.e., lipopolysaccharides/LPS) can stimulate the release of mucin by goblet cells and induce rapid reconstitution of the inner mucous layer [37]. PAMPs can also induce innate immune responses via activation of TLRs on neutrophils and macrophages [38].

Commensal bacteria can also prime dendritic cells (DCs) via their antigen presentation, which, in turn, can activate TLRs to train the innate immune system for recognition of pathogenic vs. commensal microbes [39]. Moreover, invading microbes are phagocytosed and eradicated by mucosal innate immune cells, such as DCs and macrophages in healthy conditions [40]. Of note, specific DC subsets can engulf selective bacterial species in the lamina propria at steady state [41]. It was also recently uncovered that the maturation of precursors of type 1 conventional DCs is mediated by gut-microbiota-induced tumor necrosis factor (TNF) secretion by monocytes and macrophages [42]. In addition to macrophages, neutrophils, and DCs, there are additional specialized epithelial cells, i.e., goblet cells and Paneth cells, that release various antimicrobials, such as mucins, defensins, lysozyme, secretory phospholipase A2, and cathelicidins; they serve as accessory immune cells to sustain gut innate immunity [43,44]. Innate lymphoid cells (ILCs) are another branch of the innate immune system that are mostly non-cytotoxic and secrete several effector cytokines [45]. Collectively, many innate immune cell populations maintain gut microbiota homeostasis.

In clinical illness, alterations of the enteric microenvironment promote opportunistic pathogen growth and reduce the abundance of commensal bacteria, i.e., gut microbiota dysbiosis [46], which causes imbalanced immune responses (summarized in Graphical Abstract). In a pathologic environment, neutrophils are excessively engaged into the site of inflammation or infection and can induce collateral mucosal damage via increasing pro-inflammatory cytokine secretion, matrix metalloprotease production, and pathologic immune cell activation [43,47]. Neutrophils are normally kept in a quiescent state to prevent perturbation of gut microbial ecology, which is mediated by the adapter protein downstream of kinase 3 [48]. Interestingly, induction of neutrophil extracellular traps (NETs) led to pathogen clearance and lowered inflammation [49]. Antibiotic-induced gut microbiota dysbiosis was also found to induce NETs formation, but this was associated with worsened inflammation [50], emphasizing that more investigation is needed to determine the role of intestinal NETs. Overall, an appropriate threshold or balance between the innate immune system and gut microbiota is essential to sustain homeostasis and prevent pathophysiologic outcomes.

## 4. Interaction between Gut Microbiota and Adaptive Immune System

The adaptive immune system in the gut mucosa comprises mainly intraepithelial lymphocytes (IELs) and lamina propria lymphocytes (LPLs) [51]. Among the IELs, γδ T cells are a distinct subset of T cells because they express the Helios transcription factor [52]. γδ T lymphocytes inhibit the mucosal dissemination of bacteria by secreting pro-inflammatory cytokines and antimicrobial proteins [53,54]. For example, γδ T cells stimulate CD4^+^ T cell responses, such as mucosal release of IL-22 and calprotectin [55]. Several gut bacteria species and their metabolites are noted to promote the expansion of γδ T cells, including *Desulfovibrio*-derived phosphatidylethanolamine and phosphatidylcholine [56]. Studies have shown that when intraepithelial γδ T cells are deficient, there is more bacterial translocation and expansion of invasive pathogens [57]. This is supported by diminished circulating γδ T cells in acutely septic patients [58,59] and reduced colonic γδ T cells in inflammatory bowel disease patients [60].

Interaction between the gut microbiota and adaptive immune system prevents bacterial translocation and infection (summarized in Graphical Abstract). This is supported by the finding that the gut adaptive immune system is suppressed in germ-free mice, and introduction of commensal bacteria can stimulate development of mucosal lymphocytes, e.g., CD4^+^ T cell and cytotoxic CD8^+^ T cells [61]. Both primary and secondary phases of cytotoxic CD8^+^ T cell immunity depend on CD4^+^ T cells, which require priming by professional antigen-presenting cells and are amplified by CD4^+^ T cell signaling [62]. CD8^+^ T cells eliminate intracellular pathogens (e.g., *Salmonella*), usually assisted by DC-mediated antigen presentation [63]. *Salmonella enterica serovar* Typhi can promote CD8^+^ T cells via epigenetic modification, i.e., histone methylation and acetylation [64]. Tissue resident memory CD8^+^ T cells are essential to protect against re-infection cases, and this can be studied through the Transient Microbiota Depletion-boosted Immunization model, which temporarily restrains microbiota-mediated colonization resistance [65]. Of note, B cells can also phagocytose pathogens, such as *Salmonella* and reactivate memory CD8^+^ T cells, via cross-presentation [66].

T helper 17 cells (Th17) also display distinct roles in both host protection and inflammatory responses. It appears that most Th17 responses are pathological, where one novel finding is that stem-like intestinal Th17 cells promote pathogenic effector T cells in extra-intestinal diseases [67]. Interestingly, Th17 cells stimulated by segmented filamentous bacteria (SFB) are non-inflammatory, whereas Th17 cells induced by *Citrobacter* spp. are pro-inflammatory [68]. Studies have shown that Th17 cells are absent in germ-free mice and are induced by specific microbes, such as SFB [69] and other commensal bacteria [70]. SFB-mediated IL-17 stimulation was found to be guided by cytokine (e.g., IL-6) signals [71]. The gut microbiota can also mediate Th17 responses. A study found that microbiome-dependent Th17 inflammation is regulated by α2,6-sialyl ligands, where α2,6-sialyltransferase deficiency induced mucosal Th17 responses [72]. Pathological Th17 cells can also be promoted by the *Actinobacterium Eggerthella lenta* through help by the cardiac glycoside reductase 2 enzyme [73] and *Fusobacterium nucleatum* via the short-chain fatty acid, butyrate [74].

Regulatory T cells (Treg) are another adaptive immune cell that provides immune tolerance in the GIT. Early in life, natural Treg cells are generated in the thymus via autoimmune regulator for self-tolerance [75,76], and then exposure to diet and microbiota sets in motion peripheral or inducible Treg production [32,77,78,79]. Gut microbiota can induce Treg cells by multiple mechanisms. For instance, ILCs can select for microbiota-specific RORγt^+^ Treg cells and prevent the expansion of Th17 cells to maintain immune tolerance in the intestine [80]. *Helicobacter* spp. [81] and *Akkermansia muciniphila* (*A. muciniphila*) [82] can also induce RORγt^+^ Treg cell-mediated immune responses. Comparatively, lowered levels of the gut-microbiota-derived metabolite propionate (a short chain fatty acid) can contribute to a pathological imbalance in the Th17/Treg cell differentiation [83,84].

The gut microbiota also performs a crucial role in regulating the production of secretory immunoglobulin A (IgA), which is primarily aimed against enteric commensals and bacterial antigens [85,86]. Secretory IgA can be produced either via T cell-dependent or T cell-independent pathways; T cell-dependent IgA production is more important in shaping gut microbiota homeostasis [87]. Early in life, IgA plasma cells have reactivity to commensal microbiota, which contributes to a balanced microbiome [88]. Additional evidence highlights antigenic imprinting that is essential for antibody response later in life [88,89]. This includes IgA secretion into breastmilk, where maternal transfer of IgA is imperative for immune development in the offspring [90]. When IgA is deficient, as shown in mice, gut commensals can easily cross the lamina propria, leading to enteric bacterial translocation [91].

## 5. Crosstalk between Microbial Metabolites and Immune Regulation

### 5.1. Short-Chain Fatty Acids

The gut microbiota has a huge metabolic capacity to convert host-derived and dietary components (lipids, carbohydrates, and proteins) into different metabolites that may be either favorable or dangerous for the host. Bacterial metabolites, such as short-chain fatty acids (SCFAs), secondary bile acids, lactic acid, and bacteriocins, have antimicrobial activities that protect against pathogenic bacteria [92,93]. SCFAs are produced by fermentation of indigestible carbohydrates by some commensals, including *Faecalibacterium prausnitzii, Roseburia intestinalis,* and *Anaerostipes butyraticus* [94]. SCFAs maintain intestinal homeostasis in normal colon by participating in intestinal repair through cellular proliferation and differentiation (Figure 1A). Acetate, mostly produced by *Bifidobacteria* spp., maintains gut–epithelial barrier function and regulates intestinal inflammation by activating the G-protein receptor (GPR) 43 [95]. Through GPR43 signaling, acetate promotes microbiome-reactive IgA production [96]. This relates to acetate being one of the major gut microbial metabolites to increase colonic IgA production and IgA coating on bacteria including *Enterobacterales* [97]. Acetate induction of IgA is essential to sustain gut microbiota in homeostasis. In pathophysiologic conditions, acetate and propionate, either alone or in combination, can effectively reduce inflammation by reducing Th1/Th17 and elevating T_reg_ levels [98]. Likewise, acetate supplementation to dams with preeclampsia can restore fetal thymic T_reg_ cell output [99], and acetate feeding to non-obese diabetic mice can reduce autoreactive T cells [100]. Acetate was also found to promote T cell differentiation into both effector and T_reg_ cells, which minimized *Citrobacter* infection [101].

Butyrate acts predominantly in intestinal homeostasis as an important energy source for colonocytes [95] and promotes release of mucin to maintain gut barrier homeostasis (Figure 1A) [102]. In addition to mucin, butyrate can promote the epithelial barrier through IL-10Rα-dependent repression of claudin-2 [103]. In regulating immune responses, butyrate can promote monocyte-to-macrophage differentiation by inhibiting histone deacetylase 3 (HDAC3) [104] and increasing the expression of IFN-γ and granzyme B in CD8^+^ T cells [105]. Moreover, butyrate can induce IL-22 secretion from T cells via promoting aryl hydrocarbon receptor (AhR) and hypoxia-inducible factor 1α expressions [106]. Similar to acetate, butyrate can modulate immune responses by activating GPR43 and inducing differentiation of Foxp3^+^ CD4^+^ Treg cells [100,107]. Butyrate can also promote inducible Treg production by accelerating fatty acid oxidation [108] and inhibiting HDAC [109,110]. Comparatively, the HDAC inhibitory effects of butyrate and propionate at high doses decreased class-switch DNA recombination in B cells, resulting in impairment of intestinal and systemic T-dependent and T-independent antibody responses [111]. This could explain findings from another report regarding an inverse correlation between high IgA levels and low SCFA levels that was associated with better immune tolerance [112]. Of note, in contrast to butyrate, propionate reduced IL-17 and IL-22 production by intestinal γδ T cells [113]. Overall, the main mechanisms that SCFAs maintain immune homeostasis in the intestine include HDAC inhibition, GPR signaling, inhibiting pro-inflammatory cytokine secretion, and promoting IgA production (Figure 1A).

### 5.2. Secondary Bile Acids

Bile acids are cholesterol-derived surfactants that primarily function to assimilate dietary lipids and fat-soluble vitamins. Primary bile acids (cholate and chenodeoxycholate) are produced in the liver and are secreted into the gallbladder conjugated to either taurine or glycine [114]. After traveling in the small intestine, 95% of bile acids are reabsorbed in the ileum, and the other 5% enter the colon. Conjugated cholate and chenodeoxycholate are then susceptible by the gut microbiota to a two-step bile salt hydrolase and dehydroxylation process that metabolizes them into the secondary bile acids deoxycholate (DCA) and lithocholate (LCA), respectively [115]. Secondary bile acids are regulated by intestinal clock-controlled bacteria, where a disruption in rhythmicity of the microbiota suppressed immune cell recruitment [116]. Moreover, gut microbiota regulation of secondary bile acids is apparent with the evidence that self-reinoculation (i.e., coprophagy) favored conjugated bile acids possibly due to the reduced total microbial load and low abundance of anaerobic microbiota [117]. Of note, oral administration of conjugated bile acids to newborn mice accelerated postnatal microbiota maturation [118].

Bile acids can drive metabolic and inflammatory mechanisms through activation of the nuclear receptor, Farnesoid X Receptor (FXR), or the G-protein coupled receptor, Takeda G protein-coupled receptor 5 (TGR5) [119]. Recent evidence implicates the role of secondary bile acids in both innate and adaptive immune responses. In terms of innate immunity, secondary bile acids, such as DCA, can activate TGR5, where this signaling inhibits monocyte-derived DC activation and NF-κB signaling [120] and promotes IL-10-dependent M2 macrophage polarization [121]. In relation to adaptive immunity, secondary bile acids were first noted to modulate gut RORγ^+^ Treg homeostasis, where genetic ablation of bile acid synthesis significantly depleted RORγ^+^ Treg cell counts [122]. Likewise, 3β-hydroxydeoxycholic acid (isoDCA) and isoDCA-producing Bacteroides consortia enhanced the peripheral generation of RORγt^+^ Treg cells by antagonizing FXR on DCs [123]. In addition to Treg cells, secondary bile acids, such as 3-oxoLCA and isoLCA, suppress Th17 cell function by inhibiting RORγT, a key Th17 cell-promoting transcription factor [124,125]. Unconjugated LCA also impeded Th1 activation by inhibiting ERK-1/2 phosphorylation via activation of the vitamin D receptor [126]. Furthermore, bile acid metabolism is also regulated by humoral immune responses, but dysfunction in the latter results in bile-acid-dependent small intestine enteropathy [127]. Overall, secondary bile acids modulate the gut microbiota–immune axis by lowering Th17/Treg cell differentiation, limiting pro-inflammatory cytokine secretion, and promoting M2 macrophage polarization (Figure 1B).

## 6. Influence of Environmental Microbiome Perturbation on the immune System

### 6.1. Antibiotic-Induced Microbiome Disturbances

Antibiotics have greatly improved humanity’s ability to fight infections. However, the impact of antibiotics on the microbiome was not considered until more recently. The neonatal gut microbiota and immune system can be susceptible to maternal programming when the dam microbiota is exposed to antibiotic treatment; as a result, the offspring has increased risk for developing disorders, including inflammatory bowel diseases and autoimmune diseases, and hypersensitivity, such as asthma [128,129,130,131,132,133,134]. Similarly, direct antibiotic treatment to infants, especially preterm infants, alters their microbial composition and increases susceptibility to various infections, such as necrotizing enterocolitis (NEC) [135,136,137]. It is notable that fecal microbiota transfer from NEC patients to germ-free mice demonstrated a significant reduction in butyrate and T_reg_ levels [138]. Transient antibiotic exposure to infants can also cause microbiota-dependent suppression of type 3 ILCs, resulting in late-onset sepsis [139].

Antibiotics can have several direct and indirect negative impacts on adult human health, such as the development of antibiotic resistance for select microbial species and the loss of beneficial taxa [140]. For instance, a combined administration of meropenem, gentamicin, and vancomycin increased the abundance of pathobionts, such as *Enterobacteriaceae*, and diminished butyrate-producing commensals, such as *Bifidobacterium* [141]. Similar observations were seen when oral antibiotics lowered probiotic bacteria in the microbiota [142]. It has also been reported that ciprofloxacin rapidly decreased the richness and diversity of gut microbiota accompanied by shifts in Bacteroidetes, *Lachnospiraceae*, and *Ruminococcaceae* [143].

Exposure to antibiotics affects host immune responses, and this is linked to microbiota changes. For example, a study in mice demonstrated that antibiotic-induced alterations in the microbiota shifted the Th1/Th2 balance toward Th2-dominant immunity, which reduced lymphocytes [144]. Similar findings were found in newborn macaques after early-life antibiotic exposure that rendered the animals more susceptible to bacterial pneumonia, concurrent with neutrophil senescence, hyperinflammation, and macrophage dysfunction [145]. While changes in microbial populations after antibiotic treatment vary widely [141,146], a persistent theme appears to be the short-term (and in some cases, long-term) loss of certain keystone taxa and SCFA-producing bacteria [141,147]. As emphasized in Section 5.1, SCFAs stimulate CD4^+^ T cells and ILCs to produce anti-inflammatory IL-22 by several mechanisms [80], including inhibition of HDAC and stimulation of GPR41/43 [106]. SCFAs also maintain epithelial barrier function [148]. Consistent reports demonstrate that antibiotic exposure decreases SCFA levels [149,150,151]. Overall, increase of antibiotics use in both infants and adults suggests that these complications are likely to develop more acutely or more dominant in the future. Cautious use of antibiotics and continued research into the structure and function of the gut microbiota is a prerequisite to address these challenges.

### 6.2. Fecal Microbiota Transplantation

Fecal microbiota transplantation (FMT) is a procedure in which feces are transferred from one individual to another. The goal is to restore eubiosis by introducing beneficial commensals for reversing gut microbiota dysbiosis and restoring immune function. FMT has established itself as a widely used treatment for recurrent *C. difficile* infection [152]. Recent data suggest FMT may also be effective in the treatment of type I diabetes mellitus and IBD [153,154,155,156]. Ongoing research is investigating the potential of FMT in a multitude of other disorders with established links to gut microbiota dysbiosis, including cardiometabolic syndrome, autoimmune diseases, sleep apnea, depression, and schizophrenia [157,158,159,160,161]. Several mechanisms have been suggested regarding the benefits of FMT. One example involves the Gram-negative anaerobic bacterium *Bacteroides fragilis* (*B. fragilis*). *B. fragilis* contains an extraordinary part of the genomic DNA that has been used to produce capsular polysaccharides, which are known to be central virulence factors. Among the eight capsular polysaccharides loci of *B. fragilis*, there are two capsular polysaccharides that possess a zwitterionic charge motif [162]. A recent study demonstrated that *B. fragilis* and its metabolite polysaccharide A (one of the zwitterionic polymers) have the ability to restore dysfunctional Th1/Th2 balance in germ-free mice via TLR2-mediated activation of NF-κB [163]. It is the polysaccharide’s dual-charge structural motif that confers this ability [164,165]. Another mechanistic example for FMT includes rebalancing Th17 and Treg populations as seen in colitis patients [166]. Furthermore, restoration of SCFA levels is one other mechanism of the benefits of FMT, as shown with stroke recovery [167]. As can be expected, enteral broad-spectrum antibiotics can negate the positive effects of FMT, as seen in pre-term piglets with NEC [168]. While several beneficial effects of FMT have been mentioned, it is important to acknowledge that FMT could result in the possible transfer of pathogenic microbes present in the donor feces to the transplant patient, which can cause sepsis and other diseases [8,169].

### 6.3. Diet-, Probiotic-, and Prebiotic-Induced Microbiome Alterations

The gut microbiome has a wide range of metabolic activities, including metabolizing lipids, carbohydrates, and proteins. Many recent studies have focused specifically on the link between the microbiome and diet. Dietary food additives, such as emulsifying agents, ubiquitous in highly processed foods, increase host inflammation by altering the gut microbiome [170]. On the other hand, Mediterranean style diets increase the levels of SCFA-producing bacteria and minimize inflammation [171]. In addition, low-fat vegan diets improve insulin sensitivity and body composition in obese adults by changing the prevalence of *Bacteroides* and other gut microbes [172]. Other diets, such as a high protein diet, have limited effects on microbiota composition [173]. Below, we highlight in detail other dietary sources that can have either a negative or positive impact on the gut microbiota-immune axis.

#### 6.3.1. High-Salt Diet

A high-salt diet (HSD) is associated with metabolic disorders, such as hypertension and obesity. Salt consumption greater than 20% of daily allowance is considered to be high salt intake. Salt, especially sodium, plays a crucial role in maintaining homeostasis. Sodium content in the blood regulates blood volume; higher salt increases blood volume, and, therefore, raises the blood pressure [174]. Apart from its direct effects on hemodynamics, high salt consumption can also alter the gut microbiota, which, in turn, aggravates metabolic disorders. The effect of HSD on gut microbial composition has been reported in several mouse models of various diseases [175,176,177,178]. A study by Hu et al. showed that chronic high salt intake led to enteric dysbiosis; particularly, the percentages of Actinobacteria, Firmicutes, and Bacteroidetes were markedly altered, and HSD caused gut leakiness, renal injury, and systolic blood pressure elevation [178]. Another recent study showed that administering HSD to mice for 3 weeks caused a significant increase in the Firmicutes/Bacteroidetes (F/B) ratio and Proteobacteria [179], both of which are classic markers of gut microbiota dysbiosis and are associated with metabolic disorders. Similarly, another study showed that HSD increased the F/B ratio and abundances of *Lachnospiraceae* and *Ruminococcus* but decreased the abundance of *Lactobacillus* [177]. The report by Miranda et al. further demonstrated that HSD decreases *Lactobacillus* spp. and butyrate production in a colitis mouse model [175]. In addition to microbiota changes, salt can affect immune responses. The main component of salt, i.e., sodium chloride (NaCl), induces pathogenic Th17 cells (IL-17-producing T helper cells) in both human and mouse naïve CD4^+^ T cell culture in vitro [180]. Similarly, HSD enhanced TNF-α and IL-17A in a p38-dependent manner from human lamina propria mononuclear cells [181] and stimulated intestinal Th17 responses but inhibited the function of Treg cells [182], all of which exacerbated the severity of colitis in mice. Furthermore, increased dietary salt intake upregulates Th17 cells and pro-inflammatory cytokines GM-CSF, TNF-α, and IL-2, which has made HSD an environmental risk factor for the development of autoimmune diseases [183]. Altogether, high salt intake is considered detrimental by causing negative effects on the gut microbiota and promoting pro-inflammatory mediators.

#### 6.3.2. Dietary Polyphenols

Dietary polyphenols have also been increasingly recognized for their effects on gut microbiota. These micronutrients, including, but not limited to, flavonoids, anthocyanins, catechins and tannins, can be found in a variety of foods and beverages, such as vegetables, fruits, coffee, and tea. Though only a fraction of polyphenols is absorbed in the intestines [184], a larger unabsorbed portion remains in the gut and supports the growth of select bacterial groups [185]. For example, epigallocatechin-3-gallate (EGCG; a major catechin in green tea) promotes the growth of beneficial *Bacteroides* and *Bifidobacterium* and suppresses the bloom of pathogenic *Fusobacterium, Bilophila*, and *Enterobacteriaceae* [186]. Such microbiota-modulating effects of EGCG are noted to protect against colitis [187], high-fat diet-induced obesity [188,189,190], radiation-induced mucositis [191], and *Clostridium difficile* infection (CDI) [192] in mice. Though how EGCG impacts the microbiota is not well understood, several studies propose that it could be due to the bactericidal effects of EGCG, i.e., (i) generating H2O2 that damages the bacterial cell wall [193,194], (ii) inhibiting bacterial fatty acid and folate biosynthesis [195,196], and (iii) inducing oxidative stress and reactive oxidative species (ROS) formation in susceptible bacteria [197]. Beneficial effects of polyphenols, aside from EGCG, on gut microbiota were also noted and could be referred to in a review by *Plamada* and *Vodnar* [198]. Taken together, advances in this research area help to portray tea and other polyphenol-rich foods as a new subset of prebiotics.

#### 6.3.3. Probiotics, Prebiotics, and Dietary Fiber

There is an abundance of research regarding the use of probiotics and prebiotics and studying their effects on microbiome composition. Probiotics, which often include organisms such as *Lactobacillus*, *Bifidobacteria*, and yeast, maintain integrity of the intestinal epithelial barrier by decreasing levels of LPS, protecting tight junctions, and decreasing levels of pro-inflammatory cytokines [199,200]. For a specific example, *Lactobacillus johnsonii* probiotic supplementation to dams stabilized both the maternal and offspring gut microbiota and protected pups from retroviral infection due to fewer Th2 immune responses [201]. Of note, it was recently shown that Peyer’s patches enhance and transmit probiotic (e.g., *L. reuteri*) signals to CCR6-expressing pre-germinal center-like B cells, promoting their differentiation and autocrine TGFβ-1 activation; this resulted in induction of PD-1-expressing Th1-dependent IgA, alleviation of gut microbiota dysbiosis, and protection from intestinal inflammation [202].

Prebiotics, including dietary fibers such as inulin, fructo-oligosaccharides, and galacto-oligosaccharides, selectively increase several probiotic populations, primarily *Lactobacillus* and *Bifidobacteria*. Increasing intake of dietary fiber, particularly fructans and galacto-oligosaccharides, elevated the abundance of *Bifidobacterium* and *Lactobacillus* spp. without changing the α-diversity [203]. A study has shown that when mice fed a chow diet were switched to a plant-based diet, there was a significant increase in *Bacteroides* and *Alloprevotella* and a decrease in *Porphyromonadaceae* and *Erysipelotrichaceae* [204]. Similarly, humans on a plant-based diet tend to have a higher population of *Prevotella* and are correlated with less susceptibility to gut disorders, such as IBD [2,205,206].

Both pro- and prebiotics increase SCFA levels, benefitting host immunity in various ways, including the inhibition of pro-inflammatory NF-κB pathways and induction of Treg cells [107,207]. The collective benefits of pro- and prebiotics explain their success in attenuating certain metabolic, allergic, and autoimmune diseases linked to gut microbiota dysbiosis [200,208,209,210,211]. However, it is important to acknowledge that probiotics only work when actively administered and have no proven long-term benefits. This relates to the limited knowledge about how long probiotic prophylaxis could stabilize the gut microbiota in preterm infants who are at greater risk for inflammatory diseases [212]. Albeit rarely, probiotic microbes themselves can cause bacterial infections and endotoxemia (*Lactobacillus* spp.), or negative side effects could come from a possible contamination (Mucormycetes) [8]. Similar thoughts and concerns should be applied to prebiotics as well.

## 7. Dysregulation of Microbiome–Immunity Interaction in Various Diseases

### 7.1. Gut Microbiota Dysbiosis and Immune Dysregulation

Gut epithelial cells and the mucosa serve as physical barriers against infection and endotoxemia. Gut microbiota metabolites, such as SCFA and secondary bile acids, also regulate gut permeability via immunomodulation. Of note, another gut-microbiota-derived metabolite inosine, produced by *Bifidobacterium* and *A. muciniphila*, heightens Th1 differentiation and effector function of naïve T cells [213]. Gut-microbiota-mediated immune responses are essential for preventing intestinal permeability. It is hypothesized that gut microbiota dysbiosis increases intestinal permeability from a ‘leaky gut,’ which allows opportunistic pathogens and their microbial products/toxins to invade the bloodstream and ultimately mount an inflammatory response [214,215,216]. Support for this idea comes from a number of known metabolites, such as phenolic and sulfur-containing compounds, that can harm the intestinal epithelia [217], disrupt intercellular tight junctions [218], and promote bacterial translocation [219]. These consequences, which also include immune cell dysfunction and inability to eliminate the invading pathogens, lead to inflammatory diseases [220,221]. This section of the review will discuss the microbiota-immune axis in prevalent intra- and extraintestinal diseases (Figure 2 and Table 1).

### 7.2. Gastrointestinal Infections

Depending on the context, the gut microbiota can either protect the host or increase risk of infection from exogenous pathogens. The role of the microbiome as a protective force is supported by research indicating that immature microbiomes of neonates are more susceptible to invasion by pathobionts [222]. There are several different mechanisms in which commensals can prevent colonization by pathogens and protect against infections, including competing for resources, releasing bacteriophages, and producing antimicrobial metabolites [237,238,239,240,241]. In contrast, microbiome metabolites, such as 4-methyl benzoic acid, 3,4-dimethylbenzoic acid, hexanoic acid, and heptanoic acid, have been shown to increase colonic epithelial damage, as seen by enterohemorrhagic *E. coli* in an organ-on-a-chip model [223]. Moreover, supernatant taken from commensal *Escherichia albertii* can also increase virulence of diarrheagenic *E. coli* species, resulting in a TLR5-mediated increase in IL-8 and an overall increased pro-inflammatory response by host intestinal cells [242].

Presence of certain commensals and changes in microbiome composition are linked to infection susceptibility by organisms such as *Clostridium difficile*, *Salmonella typhimurium*, *Escherichia coli*, vancomycin-resistant *Enterococcus* spp., and *Citrobacter rodentium* [238,239,241,243,244,245]. One of the best examples involves CDI, where innate immune cells are stimulated by *C. difficile*-toxins through the inflammasome and the TLR4, TLR5, and nucleotide-binding oligomerization domain-containing protein 1 (NOD1) signaling pathways [246,247]. Numerous pro-inflammatory cytokines (such as interleukin (IL)-12, IL-1β, IL-18, interferon gamma (IFN-γ), and tumor necrosis factor α (TNFα)) and chemokines (MIP-1a, MIP-2, and IL-8) are subsequently produced, resulting in increased mucosal permeability, mast cell degranulation, epithelial cell death, and neutrophilic infiltration [248]. Importantly, CDI is usually a result of antibiotic-mediated disruption of the gut microbiota [249]. Eradication of beneficial bacteria in the gut by certain antibiotics, particularly clindamycin, enables *C. difficile* to flourish [250], resulting in colitis and subsequent diarrhea [251,252]. Besides gut microbiota dysbiosis, immune cell populations, such as Th17- and IL-17-expressing cells, can promote recurrent CDI [253]. Comparatively, IL-33-activated ILCs can prevent CDI [254]. As gut microbiota depletion is a main cause for CDI, interventions that restore microbes could be of therapeutic value.

Prebiotics, such as dietary fiber and their fermented byproducts, i.e., SCFA, are possible treatments for CDI. For instance, dietary fibers, such as pectin, were able to restore gut microbiota eubiosis (denoted by increased *Lachnospiraceae* and decreased *Enterobacteriaceae*) and alleviate inflammation following *C. difficile*-induced colitis [255]. The butyrate producing bacterium *Clostridium butyricum* was similarly found to protect against CDI by increasing neutrophils, Th1, and Th17 cells in the early phase of infection; this was independent of GPR43 and GPR109a signaling [256]. As mentioned in Section 6.2, CDI can be effectively treated by FMT [152]. FMT is further supported in a prior study that showed that a Microbial Ecosystem Therapeutic, consisting of 33 bacterial strains isolated from human stool, could treat antibiotic-resistant *C. difficile* colitis [257]. Of note, similar observations were seen when the Microbial Ecosystem Therapeutic was applied to *Salmonella typhimurium* infection [258]. These findings emphasize that appropriate modulation of the gut microbiota and immune responses are imperative for preventing and fighting against infection.

### 7.3. Inflammatory Bowel Diseases

Inflammatory bowel diseases (IBD) develop due to defects in various factors, such as environment, gut microbes, immune system, and genetic factors. IBD involves chronic inflammation of the GIT. Crohn’s disease (CD) and ulcerative colitis (UC) are two distinct clinical conditions of IBD based on histopathological features, location of disease in the GIT, and symptoms [259]. In IBD, mucolytic bacteria and pathogenic bacteria degrade the mucosal barrier and increase the invasion of pathogens into deep intestinal tissues [224,260,261,262]. Alterations in the gut microbiota composition have been highly linked to the development and progression of IBD. IBD patients show reduced populations of Firmicutes and an expansion of Proteobacteria, Bacteroidetes, *Enterobacteriaceae*, and *Bilophila* [263,264,265]. In addition, many pro-inflammatory bacterial species are coated with IgA, as seen in IBD patients and colitis mouse models [266,267]. Gut microbes appear to play a direct role in IBD development on the basis of the evidence that germ-free mice are protected against colitis [268]. This is reinforced by the discovery that implantation of gut microbes from IBD mice to germ-free mice resulted in IBD for the latter group [268]. Likewise, dams with IBD can essentially transfer an ‘IBD microbiota’ to the offspring, for which the pups have reduced microbial diversity and fewer class-switched memory B cells and Treg cells in the colon [269]. The strong link between microbiota and IBD has moved forward metagenomic approaches to help better identify diagnostic and therapeutic targets [270].

FMT is proposed as a potential treatment, where treated UC patients were found to have an increased abundance of *Faecalibaterium* that corresponded with less RORγt^+^ Th17 cells and more Foxp3^+^ CD4^+^ Treg cells [166]. Administration of SCFAs is also thought to be a potential therapeutic for IBD patients [271]. Supporting evidence includes butyrate-mediated inhibition of pro-inflammatory neutrophil responses, i.e., NETs in colitic mice [272]. There are conflicting reports as to whether dietary fiber, the precursor for SCFA, could be a beneficial intervention for IBD patients. On one side, a specific multi-fiber mix was found to counteract intestinal inflammation via increasing IL-10 and Treg cells [273]. Opposingly, our research findings indicate a dichotomy in prebiotic fiber reactions for colitic mice, where pectin could alleviate inflammation compared with inulin, which aggravated the disease pathology [274]. Moreover, our study suggested that butyrate could be a detrimental microbial metabolite by increasing NLRP3 inflammatory signaling [274]. A probiotic cocktail, comparatively, alleviated inflammation by shifting the gut microbiota to an anti-inflammatory profile which included *Akkermansia* and *Bifidobacterium* [275]. These findings collectively indicate that more investigation is required to understand prebiotic fibers and SCFAs in IBD before implementing it in the clinics.

In addition to SCFA, secondary bile acids are implicated in IBD. DCA has been well-established to induce intestinal inflammation [276,277]. This could be due, in part, to bile-acid-mediated inhibition of Paneth cell function [278]. Yet, cholecystectomy-associated secondary bile acids, including DCA, ameliorated colitis in mice by inhibiting monocytes/macrophages recruitment [279]. Moreover, UDCA can also lower colitis severity by preventing the loss of *Clostridium cluster XIVa* and increasing the abundance of *A. muciniphila* [280]. The varying effects of bile acids could be related to their chemical structure and potential conjugated moieties. For instance, sulphated secondary bile acids may exert more pro-inflammatory effects compared with their unconjugated counterparts, as seen in IBD patients [281]. Certainly, more metabolomic profiling is necessary to understand the bile acid profile in IBD patients and determine the pro- or anti-inflammatory effects for each type of bile acid. In general, it appears that both SCFA and secondary bile acids have anti-inflammatory effects in the intestine (Figure 1A,B).

Several susceptibility genes that increase risk for IBD have been identified in recent years. Current research is focused on the idea that genetic predisposition, dysbiosis, and environmental factors, such as antibiotics, work in concert toward IBD. Nucleotide-binding oligomerization domain-containing protein 2 (NOD2, an immunological intracellular recognition protein) identifies intracellular muramyl dipeptide (MDP), an integral component of bacterial cell walls [282]. Loss of NOD2 function impairs inhibition of TLR2-mediated activation of NF-κB, resulting in an overactive Th1 response and weakened immunological tolerance to microbes [282]. Moreover, several other genes that increase susceptibility to IBD, including autophagy-related 16-like 1 (ATG16L1), caspase recruitment domain-containing protein 9 (Card9), and C-type lectin domain family 7 member A (CLEC7A), dysregulate T cell responses and create gut microbiota dysbiosis, also contributing to IBD [283,284,285]. Future studies should explore whether there are single nucleotide polymorphisms in genes related to microbial metabolite production for IBD patients.

### 7.4. Colorectal Carcinoma (CRC)

A growing body of literature suggests a role for microbiota in the development and progression of cancer. In scenarios where the immune system has maladaptive development, gut microbiota dysbiosis becomes a high risk, and the expansion of certain microbes can result in the production of mutagenic toxins [286]. These genotoxins include *Bacteroides fragilis* toxin (Bft), cytolethal distending toxin (CDT), and colibactin [225]. However, these highlight only a small number of bacterial-related toxins where more research is needed to identify and understand the carcinogenic potential with the full breadth of gut microbes [225].

Adenomatous and serrated polyps are two precancerous lesions that often progress to colorectal cancer (CRC). In patients with adenomas, several species, including *Bilophila*, *Desulfovibrio*, *Mogibacterium*, and the phylum Bacteroidetes, are increased in the feces, while patients with serrated polyps showed increases in the taxa Fusobacteria and class Erysipelotrichia [226]. *Fusobacterium nucleatum* (*F. nucleatum*) is characterized as an important microbe in CRC progression [287,288]. *F. nucleatum* promotes TLR4 signaling and E-cadherin/β-catenin signaling, ultimately leading to activation of NF-κB and reduced miR-1322 expression [289]. Regulatory micro-RNAs, such as miR-1322, can directly regulate the expression of CCL20, a cytokine that promotes CRC metastasis [287]. Other literature points to *F. nucleatum* adhesin A (FadA) as a key virulence factor that allows *F. nucleatum* to adhere, invade, and erode the colonic epithelia [227]. More recently, one study found that *F. nucleatum* can promote CRC by suppressing anti-tumor immunity through activation of the inhibitory receptors CEACAM1 and TIGIT1, which downregulate NK cells and T cells [290]. The *F. nucleatum* strain Fn7-1 was also demonstrated to aggravate CRC development by elevating Th17 responses [74]. These findings on *F. nucleatum* are alarming because this is a SCFA-producing bacterium [291], and SCFA have been, in general, highlighted as a potential therapeutic avenue for many inflammatory diseases. *F. nucleatum* predominantly produce acetate and butyrate, where it was recently suggested that *F. nucleatum* induces Th17 via free fatty acid receptor 2 (FFAR2), a SCFA receptor [74]. Yet, loss of FFAR2 in mice aggravated tumor bacterial load and over activated DCs, eventually promoting T cell exhaustion [292]. Moreover, butyrate from dietary fiber was found to be less metabolized in CRC cells because of the Warburg effect, allowing it to act as an HDAC inhibitor and promote acetylation of genes related to apoptosis [293]. These findings emphasize that the pathologic effects of *F. nucleatum* could be SCFA-independent, but further studies are needed to determine this possibility.

Another proposed mechanism in the development of CRC suggests that excessive dietary intake of sugars, proteins, and lipids could promote the growth of bile-tolerant microbes, which increase production of secondary bile acids, such as DCA and LCA, and by-products, such as hydrogen sulfide. Excessive secondary bile acids are genotoxic and may produce a pro-inflammatory environment that could promote the development of CRC [226]. In particular, DCA can stimulate intestinal carcinogenesis by activating epidermal growth factor receptor-dependent release of the metalloprotease ADAM-17 [294]. DCA also activates β-catenin signaling [295] and drives malignant transformations in Lgr5-expressing (Lgr5+) cancer stem cells [296] for CRC growth and invasiveness. However, bacteria associated with secondary bile acid production, i.e., *Clostridium cluster XlVa*, were significantly decreased in IBD patients, which was accompanied by reduced transformation of primary to secondary bile acids [297]. In addition to bile acids, the gut microbial metabolite folate can worsen CRC pathogenesis by triggering AhR signaling and expanding Th17 levels [298]. Similar to SCFA, more investigation is needed to discern the potential pro-tumorigenic effects of gut-microbiota-derived bile acids.

There are distinct microbiota-dependent immunological responses in CRC. In terms of innate immune responses, *A. muciniphila* enrichment facilitated M1 macrophage polarization in an NLRP3-dependent manner that suppressed colon tumorigenesis [299]. Likewise, intestinal adherent *E. coli* can increase IL-10-producing macrophages, which limits intestinal inflammation and restricts tumor formation [300]. In terms of adaptive immunity, microbial dysbiosis hyperstimulates CD8^+^ T cells to promote chronic inflammation and early T cell exhaustion, which contributes to colon tumor susceptibility [301]. Intestinal cancer cells can also respond to the microbiota by inducing calcineurin-dependent IL-6 secretion, which promotes tumor expression of the co-inhibitory molecules B7H3/B7H4 that diminish anti-tumor CD8^+^ T cells [302]. Comparatively, introduction of *Helicobacter hepaticus* induced T follicular helper cells that restored anti-tumor immunity in a mouse CRC model [303]. Compared with macrophages and Th17 cells, γδ T cells and resident memory T cells were found at lower frequencies in the colonic tissue of CRC patients [60]. It would be interesting to investigate whether an immune cell panel could be developed for early diagnosis of CRC.

### 7.5. Hepatocellular Carcinoma (HCC)

Hepatocellular carcinoma (HCC), the most common primary liver cancer, is the fourth leading cause of cancer-related mortality worldwide [304]. The main etiology for HCC pathogenesis comes from pre-existing liver diseases, such as nonalcoholic fatty liver disease (NAFLD) and steatohepatitis, that lead to cirrhosis [305]. This is further complicated by other concomitants in NAFLD patients, including insulin resistance, obesity, and metabolic disorders that further promote hepatic inflammation and tumorigenesis through IL-6 and TNF-α [306]. The liver is the ‘first stop’ for venous blood coming from the intestines, making it vulnerable to the gut microbiota via microbial translocation across the intestinal–epithelial barrier or contact with absorbed microbial metabolites [307]. The aforementioned well-known effects of gut microbiota dysbiosis, including disruption of gut barrier, translocation of microbes into the bloodstream, and subsequent inflammatory immune responses via induction of PRRs by PAMPs, such as LPS, are strongly correlated to the pathogenesis of NAFLD, liver cirrhosis, and HCC [228,307]. While it has long been thought that gut microbiota dysbiosis precedes the development of HCC, this causal relationship has not been explored in depth until more recently. Behary, Raposo et al. recently found, before HCC progression, that gut microbiota dysbiosis is in tandem with early onset liver injury that is followed by an LPS-dependent Th1- and Th17-mediated cytokine response [308]. Further investigation should determine whether gut microbiota dysbiosis is a cause or consequence in the liver injury preceding HCC.

Increased *Enterobacteriaceae* and *Streptococcus* and reduction in *Akkermansia*, alongside elevated levels of inflammatory mediators, such as CCL3, CCL4, CCL5, IL-8, and IL-13, have been noted in patients with NAFLD-associated HCC [309]. A more recent study found decreased abundance of SCFA-producing bacteria and increased LPS-producing bacteria in patients with cirrhosis-induced HCC but no significant evidence of gut microbiota dysbiosis in other liver diseases, such as hepatitis C, hepatitis B, or alcoholic liver disease [310]. Broadly speaking, however, it should be noted that altered microbial populations observed among multiple studies are not consistent with each other [309,311,312,313]. Furthermore, while it is generally thought that SCFAs produced by gut microbes have several benefits for humans, it was recently discovered that inulin, a precursor of the SCFA butyrate, can promote the progression to HCC in genetically altered dysbiotic mice [229]. Other studies have focused on the impact of microbial metabolites on HCC. For instance, a high-fat diet led to gut overgrowth of Gram-positive organisms that generate secondary bile acids, i.e., DCA [230]. DCA can work in concert with lipoteichoic acid to activate TLR2 and subsequently downregulate anti-tumor immunity, creating a microenvironment favorable for the development of HCC [314,315]. Overall, it appears that gut microbiota metabolites are potentially pro-tumorigenic for the liver.

### 7.6. Cardiovascular Disease

Cardiovascular disease (CVD) is heavily linked to metabolic syndrome, a condition which involves a set of interrelated diseases—mainly atherosclerosis, NAFLD, hypertension, and type II diabetes mellitus (TIIDM)—that arise from chronic, low-grade inflammation [316]. Many cells with high metabolic activity, such as parenchymal cells in the liver and pancreas, adipocytes, and skeletal myocytes, participate in extensive crosstalk with immune cells. Any perturbation of the microbiome has the potential to alter host immune function and, by extension, may have the ability to cause or alter disease processes in metabolically active tissues. The recognition of LPS and other microbial PAMPs by PRRs are thought to be a key driver in this low-grade inflammatory state [231]. Trimethylamine-N-oxide (TMAO), a microbial co-metabolite, is also noted to cause low-grade inflammation through NF-κB signaling, inflammasome activation, and increased production of free radicals [317,318]. Furthermore, TMAO leads to atherosclerosis and, thus, heart disease by impairing cholesterol metabolism in macrophages and contributing to the formation of foam cells [319]. Indeed, higher serum TMAO is correlated with increased risk of atherosclerosis, coronary artery disease, stroke, and vascular inflammation [232,233], and TMAO is currently being considered as a biomarker for adverse cardiovascular events [320]. More recent research has discovered phenylacetylglutamine (PAGln) as a microbial metabolite related to CVD via adrenergic receptor activation and pro-thrombotic effects [321,322]. There are multiple potential emerging roles for PAGln in cardiovascular medicine, such as being used as a diagnostic marker or even as a predictor for responsiveness to β-blocker therapy for CVD patients [322].

### 7.7. Diabetes

Diabetes mellitus is a disease separated into two classes: type I diabetes mellitus (TIDM) involves autoimmune destruction of pancreatic islet cells, while type II diabetes mellitus (TIIDM) involves acquired insulin insensitivity. Though much research involving microbiota and diabetes revolves around TIIDM and obesity, it has been shown that increasing dietary SCFA consumption can lead to altered microbiota and distinct immune profiles in TIDM patients [323]. Increasing dietary SCFAs, such as butyrate and acetate, were also shown to work synergistically to confer protection against autoreactive T cell populations and TIDM in mice [100]. Comparatively, administration of *Parabacteroides distasonis* accelerated the development of T1DM in a mouse model, and this was because of aberrant immune responses, including elevated CD8^+^ T cells and decreased Foxp3^+^ CD4^+^ Treg cells [324]. Of note, dysregulated bile acid metabolism was found to be a potential predisposing factor for islet autoimmunity and type 1 diabetes [325].

The microbiome and immune systems are both heavily involved in the pathogenesis of TIIDM. Branched-chain amino acids are produced by *Prevotella copri* (*P. copri*) and *Bacteroides vulgatus* spp., and *P. copri* directly induces insulin resistance in mouse models [326,327]. Depletion of commensal *A. muciniphila* compromises the intestinal barrier, resulting in translocation of endotoxin into the bloodstream and subsequent activation of CCR2+ monocytes. This results in conversion of pancreatic B1a cells into 4BL cells, which release inflammatory mediators and cause reversible or irreversible insulin resistance [328]. On the other hand, microbial metabolites, such as linoleic acid and docosahexaenoic acid, have protective effects against insulin resistance and TIIDM through anti-inflammatory effects and prevention of lipotoxicity [329]. FMT has also been shown to reduce fasting blood glucose levels and decrease insulin resistance in mice with TIIDM [330]. Furthermore, some of the therapeutic effects of several anti-diabetic drugs can be due, in part, to their ability to alter the microbiota [331,332,333].

### 7.8. Hypertension

Several studies have observed significantly altered microbiome compositions between normotensive and hypertensive mice, though specific microbial profiles in hypertensive mice are dependent on the hypertension model used [334,335,336,337]. In the angiotensin II model of hypertension, lack of microbiota in germ-free mice protected against hypertension partly by decreasing inflammatory cell populations in the blood [338]. Yet, germ-free mice were more prone to kidney injury following an angiotensin II and high-salt diet combination regimen [339]. Furthermore, reintroduction of microbiota to hypotensive germ-free mice re-established vascular contractility [340]. Generally, the microbiota composition differs between hypertensive and normotensive animals and, interestingly, cross-fostering hypertensive pups with normotensive dams can reduce blood pressure in the former group [341]. Similar to CVD, the gut metabolite TMAO also has relevance to hypertension. A recent study discovered TMAO exacerbated vasoconstriction via ROS in angiotensin II-induced hypertensive mice [342]. In a similar manner, high-salt-induced DC activation is associated with microbial dysbiosis-mediated hypertension [343]. Comparatively, the ketone body β-hydroxybutyrate has been shown to be decreased in high-salt-fed hypertensive rats; rescuing with the β-hydroxybutyrate precursor 1,3 butanediol decreased blood pressure and kidney inflammation through prevention of the NLRP3-mediated inflammasome [344]. While HSD has been shown elsewhere to decrease *Lactobacillus* spp. and induce Th17 cell populations, this appears to be through a distinctly different mechanism [176].

### 7.9. Rheumatoid Arthritis

The pathogenesis of rheumatoid arthritis (RA), a systemic autoimmune disease characterized primarily by inflammation of joints, is becoming more understood. RA is a multifactorial disease with multiple identified alleles and environmental factors conferring increased susceptibility to the disease. A potentially important microbial genus in the development of RA is *Prevotella*. This was first identified in 2013 by Scher et al., which found that patients with new onset RA had significantly increased abundance of *Prevotella* spp., particularly *Prevotella copri*, compared with healthy controls [234]. However, the *Prevotella* population did not increase in patients with chronic RA [234]. Since then, multiple studies have found further correlations between various *Prevotella* species and RA [345,346,347]. However, it is unclear whether *Prevotella* spp. itself contributes to the pathogenesis of RA, or the immunological environment created by RA increases abundance of *Prevotella* in the gut.

Other notable bacterial shifts in the gut microbiota for RA patients include a bloom in Proteobacteria, *Clostridium cluster XlVa*, and *Ruminococcus*, which were correlated with less CD4^+^ T cells and Treg cells [348]. Using the K/BxN autoimmune arthritis model, it was found that SFB-mediated cytotoxic T lymphocyte antigen-4 (CTLA-4) reduction caused autoreactive T follicular helper cells [349,350]. The accumulation of T follicular helper cells and Th17 cells in arthritis appears to be age-dependent [351], which helps to explain why RA is found mostly in the older population. Interestingly, though, the gut microbiota seems to predominantly affect T follicular helper cells, not Th17 cells, as confirmed by antibiotic treatment of the K/BxN autoimmune arthritis model [352]. Of note, it was recently reported that collagen-induced RA in mice causes an aberrancy in circadian rhythmic patterns in the gut microbiome, resulting in reduced barrier integrity due to an alteration in circulating microbial-derived factors, such as tryptophan metabolites [353].

SCFAs, specifically butyrate, have been proposed as a therapeutic option for RA. Butyrate supplementation was found to promote Treg cells by inhibiting HDAC expression, and it downregulated pro-inflammatory cytokine genes in RA [354]. Moreover, butyrate alleviated arthritis by directly inducing the differentiation of functional follicular Treg cells in vitro by enhancing histone acetylation via HDAC inhibition [355]. Furthermore, butyrate reduced arthritis severity by increasing the levels of AhR ligands, i.e., serotonin-derived metabolite 5-hydroxyindole-3-acetic acid, where AhR activation supported regulatory B cell function [356]. In addition to SCFA, the gut-microbiota-derived metabolites LCA, DCA, isoLCA, and 3-oxoLCA were also very recently found to exhibit anti-arthritis effects. Specifically, isoLCA and 3-oxoLCA inhibited Th17 differentiation and promoted M2 macrophage polarization [357]. These effects of secondary bile acids could be synergized with *Parabacteroides distasonis* probiotic supplementation [357]. The newfound findings of secondary bile acids are monumental and need additional investigation.

### 7.10. Allergic Diseases

Allergies occur when the immune system becomes hypersensitized to nonpathogenic foreign antigens. Common hypersensitivities include allergic rhinitis, food allergy, eczema, atopic dermatitis, and asthma. Several factors responsible for the development of allergies, such as reduced microbial exposure, cesarean delivery, diet, and antibiotic use are strongly linked to changes in gut microbiome composition [358,359,360,361]. Gut microbiota dysbiosis, in turn, increases risk for allergies, particularly food allergies [235,236]. Dysbiosis induced by antibiotic use is sufficient to increase allergic symptoms, elevate intestinal inflammation, and disrupt gut mucosal tight junction in sensitized mice [362]. A high-fat diet generally has effects similar to antibiotics, causing gut microbiota dysbiosis and subsequently increasing risk for food allergies [363]. Changes in gut microbiota composition immediately after birth, when the microbiome is still establishing, appears to have a particularly large impact on the development of allergic diseases later in life [364]. Of note, the vaginal microbiota can also reflect allergy risk, where *Lactobacillus*-dominated vaginal microbiota clusters were related to infant serum IgE status at 1 year of age [365].

Several studies reinforce the concept that dysbiosis is heavily linked to allergic disease, especially asthma. Individuals with atopic asthma have significantly higher fecal levels of *Lactobacillus* and *E. coli* compared with healthy individuals [366]. In terms of microbiota metabolites, 12,13-diHOME (a relatively uncharacterized linoleic acid) is commonly found in neonates at high risk for asthma [367]. It was recently found that the bacterial epoxide hydrolase, which produces 12,13-diHOME, is also higher in concentration during pulmonary inflammation, and 12,13-diHOME reduced Treg cells in the lung [368,369]. Comparatively, the AhR ligand tetrachlorodibenzo-p-dioxin was able to attenuate delayed-type hypersensitivity by inducing Treg cells, suppressing Th17 cells, and reversing gut microbiota dysbiosis [370]. Likewise, individuals with higher fecal SCFAs, such as butyrate and propionate, early in life had markedly decreased risk for development of asthma and atopy [371]. Of potential therapeutic value, SCFA supplementation could modulate T cells and DCs to alleviate asthma [372]. Similarly, maternal supplementation with dietary fiber or acetate was shown to protect neonates from asthma by promoting acetylation of the *Foxp3* gene [373]. Dietary fiber feeding also gave protection from food allergens via retinal dehydrogenase activity in CD103^+^ DCs [374]. Of note, the dietary fiber inulin was recently found to promote allergen- and helminth-induced type 2 inflammation, and this was bile-acid dependent [375]. Overall, it appears that the influence of gut microbiota on allergies is highly regulated by metabolites, but each microbial product has independent effects that can either promote or demote hypersensitivity.

### 7.11. Psychiatric Disorders: The Gut–Brain Axis

The aforementioned information describes the gut microbiota to influence both intra- and extraintestinal diseases. One other organ that the gut microbiota can impact is the brain where a ‘stressed gut’ is becoming more recognized as a pathologic entity in several neurological disorders. For pre-term infants with an immature gut microbiota, *Klebsiella* overgrowth has been found to be highly predictive for brain damage and is associated with a pro-inflammatory immunological tone [376]. Parkinson’s disease is marked by an accumulation of alpha-synuclein in the gut, and patients often suffer from a leaky gut due to microbiota dysbiosis with higher populations of *Prevotellaceae* [13]. These symptoms can be reversed by administering probiotics [377,378]. Recently, the idea that microbiota shapes mental health has started gaining traction. Taxonomic and metabolic signatures have been proposed as a biomarker for stratifying major depressive disorder into mild, moderate, and severe symptom categories [379]. Several studies studying differences in microbiota between those who are mentally healthy and those with mental health disorders, such as anxiety and/or depression, have suggested that microbial colonization before and after birth plays a major role later in life. For instance, maternal stress can induce abnormal neurodevelopment in the offspring, which has been marked with a significant reduction of *Bifidobacterium* spp. [380]. Moreover, neonates delivered by C-section, as opposed to vaginal birth, have a greater risk of developing psychosis later in life [377,381]. Impressively, early-life oxytocin treatment can minimize behavior deficits seen in C-section delivered pups [382].

A cocktail of broad-spectrum, gut-microbiota-depleting antibiotics, specifically at the postnatal and weaning stages, can cause long-lasting effects of anxiety-related behavioral outcomes into adolescence and adulthood [383]. A recent elegant study by Li et al. delineated that infant exposure to antibiotics resulted in anxiety- and depression-like behaviors and memory impairments that were concurrent with an increase inflammatory milieu; similar findings were seen following long-term antibiotic treatment at the adolescent and adult stages in mice [384]. Early-life disruption of the gut microbiota could also cause sex-specific anxiety-like behavior, where LPS treatment to Wistar rats resulted in less social interaction in males compared with the females, who had an increase in social behavior [385]. It is noteworthy that FMT from an ‘aged microbiome’ to germ-free mice decreased SCFAs, and this was associated with cognitive decline [386]. The gut microbiota–immunity–brain axis is still in its nascency and requires investigation to establish mechanisms involved in immune regulation responsible for behavioral abnormalities and neurological disorders. However, it must be emphasized to look at other microorganisms besides bacteria because mucosal fungi were found to promote social behavior through complementary Th17 immune mechanisms [387].

## 8. Relationship between the Gut Microbiota and Their Metabolites in Immunotherapy

Presently, frontline immunotherapy treatments include T cells (checkpoint inhibitors, costimulatory receptor agonists), T cell modification, adoptive T cell transfer, autologous cytokine-induced killer cells, chimeric antigen receptor therapy, cytokines, oncolytic viruses, and vaccines [388,389]. In recent years, immunotherapy based on the application of immune checkpoint inhibitors (ICIs), including antibodies against CTLA-4, programmed cell death protein 1 (PD-1), and programmed death ligand 1 (PD-L1), has been approved as first- or second-line treatments in a variety of tumors [390]. In particular, ICIs that target PD-1 and its ligand PD-L1 have been approved by the U.S. Food and Drug Administration (FDA) for the treatment of 10 different cancer types [391]. Recent studies suggest the gut microbiota could be a significant determinant of the response to cancer immunotherapy in some preclinical and clinical studies [392,393,394]. Matson et al. showed that *Bifidobacterium longum*, *Collinsella aerofaciens*, and *Enterococcus faecium* have higher abundance in patients responding to PD-1 inhibitors [395]. Several studies have found significant differences in the microbiomes of responders vs. non-responders to PD-1 inhibitors, including increases in *Faecalibacterium*, *Ruminococcus,* and *Akkermansia* in responders and increases in *Bacteroides* in non-responders [392,396,397]. In addition, anti-PD-1 treatment for liver cancer patients resulted in elevated *Faecalibacterium* abundance and better progression-free survival [398].

Additional studies have further shown that the composition of gut bacteria can influence the metabolism of certain immunotherapeutic drugs. The fecal transfer from PD-1-treated responding patients to germ-free mice enhanced T cell responses and improved the effectiveness of PD-1 inhibitor therapy [395]. Inosine, which is produced by *Bifidobacterium pseudolongum* and *Akkermansia muciniphila*, also promoted anti-CTLA-4 and anti-PD-L1 therapy by activating T cells [213]. Comparatively, a recent study by Coutzac et al. showed that butyrate and propionate limited the efficacy of CTLA-4 inhibitors, which was associated with a higher Treg population and lower survival [399]. Of note, a newly isolated probiotic *Lactobacillus* strain (*L. paracasei sh2020*) promoted anti-PD-1 effects in CRC tumor-bearing mice by upregulating the expression of CXCL10 in the tumors and subsequently enhancing CD8^+^ T cell recruitment [400]. Remarkably, these anti-tumor effects occurred even in the presence of gut microbiota dysbiosis. These preclinical and clinical pieces of evidence support continued investigation to determine the requirement for gut microbiota to provide the maximum efficacy of immunotherapies (Figure 3). This includes possibly utilizing the gut microbiota to limit negative side effects from immunotherapies, such as ICI-related cardiotoxicity. Chen et al. elegantly described PD-1/PD-L1 inhibitor to deplete the *Prevotellaceae* and *Rikenellaceae* microbiota populations, reduce butyrate levels, and promote pro-inflammatory macrophage M1 polarization via downregulation of the PPARα-CYP4 × 1 axis [401]. Of therapeutic relevance, *Prevotella loescheii* recolonization and butyrate supplementation alleviated PD-1/PD-L1 inhibitor-related cardiotoxicity [401]. As immune checkpoints are often heterogeneous and not persistent, which can result in lower treatment response rate, drug resistance, and adverse reactions [402,403,404], gut-microbiota-targeted therapies could be essential adjuvants (Figure 3).

The currently approved and available IBD therapies are anti-TNF agents, anti-integrin agents, anti-β7 monoclonal antibody, and Janus kinase (JAK) inhibitors. JAK inhibitors (e.g., baricitinib) were successful in restoring insulin signaling and improving myosteatosis following high-fat–high-sugar feeding, but it did not reverse diet-induced alterations to the gut microbiota in mice [405]. Anti-TNF inhibitors have improved clinical outcomes in both CD and UC, but they still require more randomized clinical trials [402]. However, it is notable that FMT was recently found to be a potential alternative therapy for CD patients with prior loss of response or intolerance to anti-TNF therapy (i.e., infliximab) [406]. Impressively, the probiotic *Bifidobacterium longum* (*B. longum*) CECT 7894 promoted infliximab efficacy in a mouse colitis model by reducing the abundance of opportunistic pathogens, i.e., *Enterococcus* and *Pseudomonas*, and increasing secondary bile acids [407]. Another recent study similarly found that both anti-TNF and anti-IL-12/23 therapies altered the gut microbiota to favor microbial species capable of secondary bile acid production [408]. The elevation in secondary bile acids may be due to anti-TNF treatment promoting the bloom of *Clostridia* spp. as part of the restoration of intestinal microbiota [409]. Bile acids are considered to be a potential metabolic biomarker for anti-TNF therapy response [410], but more research is needed to determine whether bile acids improve immunotherapy efficacy (Figure 3). There is a hint that secondary bile acids could be beneficial when considering the evidence for UDCA treatment to prevent CRC reoccurrence by inhibiting NF-κB signaling [411,412]. Moreover, UDCA was found to synergize with anti-PD1 effects to inhibit cancer progression in tumor-bearing mice [413]. Overall, it appears that the gut microbiota could be exploited as both a biomarker and therapeutic target to improve immunotherapy response.

## 9. Promises, Challenges, and Risks in Immune–Microbiome Research

The interplay between the microbiota and immune systems and their impact on diseases, including IBD, autoimmune arthritis, and cancer, is incredibly complex. One layer of complexity includes the challenge of showing the exact implication of a certain single or group of bacteria in the onset of disease or general host physiology. Colonization of microbes to germ-free models is a relevant strategy to better understand the potential effects of gut microorganisms in host health and disease [414]. However, the gut microbiota is much more than just a select few species. There is a strong dynamic in the microbiome environment, where species are either mutually exclusive or competitive for resources, and many microbes depend on one another for growth [415]. Another layer of complexity is including other interacting genetic and environmental factors, such as diet, smoking, drugs, and medications (Figure 2). This includes differences in the microbiota (and potentially immune responses) between urban vs. rural areas for individuals [416]. Notwithstanding, observations seen in rodent models is not always translatable to humans. It can be generally stated that humans and other mammals live in a ‘dirtier’ environment when compared with research rodents living in specific pathogen-free environments. Therefore, the cleanliness of the environment, reflecting the hygiene hypothesis, could impact the microbiota composition and disease susceptibility. This notion is supported with the recent finding that feralized mice (animals continuously exposed to a livestock farmyard-type environment) had a more stable gut microbiota and remained resistant to mutagen- and colitis-induced neoplasia when compared with hygienically born mice [417].

Several studies focusing on microbiome–immunity research have employed 16S rRNA sequencing to characterize the microbiome, but this method has limitations in that it can successfully identify genera but cannot provide distinctions at the species level [418]. Therefore, to achieve a more inclusive study of microbiomes, it is advisable that metagenomics must be combined with other -omics approaches [419]. Most recently, metatranscriptomics and metabolomics are rapidly becoming important to microbiome studies. Metagenomics generates the taxonomical profile of the sample, metatranscriptomics obtains a functional profile, and metabolomics finalizes the depiction by determining which byproducts are released by the microbiota in the environment [419]. Though each of these -omics approaches provide valuable information by themselves, it is suggested that a more complete picture come from combined -omics. One important benefit with these -omics approaches is that the raw files can be deposited into databases and then later mined for analysis by other research groups. One limitation that can arise when applying machine learning to compare multiple databases is the unevenness in sample size [420]. Moreover, -omics results could be considered study-specific, as it can be difficult to find overlapping patterns of gut microbiota changes between research and/or clinical studies. This is because the gut microbiota (plus their metabolites) and disease susceptibility can vary in humans depending on their geographic origin [421], and even the bacterial composition in common laboratory rodents can be different among research facilities and vendors [422]. Overall, -omics are surely advancing the biomedical field to identify potential diagnostic and therapeutic targets, but there are still some limitations to overcome.

## 10. Conclusions

In summary, the host immune system and the gut microbiome are heavily dependent upon each other for normal function and well-being of the host (summarized in Graphical Abstract). This review covered novel findings, including how fetal immune fitness is environmentally dependent on the maternal microbiota (healthy vs. dysbiosis or stressed). New mechanistic pathways have been discussed, such as SCFA and secondary bile acids modulating gut homeostasis by inducing Treg cells and IL-10 secretion (Figure 1A,B). Throughout the review, butyrate and its precursor dietary fiber were repeatedly mentioned to influence immune responses and act as potential therapeutics for many diseases, but some evidence suggest that their clinical practice may need to be disease contextualized. Comparatively, probiotics and FMT look to be more promising to restore gut microbiota eubiosis and alleviate inflammatory diseases. Moreover, gut microbiota appears to be a relevant target to improve current immunotherapies and abate their negative side effects (Figure 3). We also discussed the current challenges in microbiome research, which is essentially rooted in genetic and environmental factors (Figure 2), that makes each individual microbiota unique among humans and when comparing species models. We posit that recent developments in multi-omics methods, including epigenomics, meta-genomics, meta-proteomics, metabolomics, culturomics, and single-cell transcriptomics, will elucidate interactions between the gut microbiome and the immune system in health and disease [423]. As such, it will be exciting to predict the ‘specific’ host immune responses on the basis of gut microbiome profiles, which will support the development of ‘personalized microbiome-targeted’ therapy for immunologic diseases.

## Figures and Tables

**Figure 1 biomedicines-11-00294-f001:**
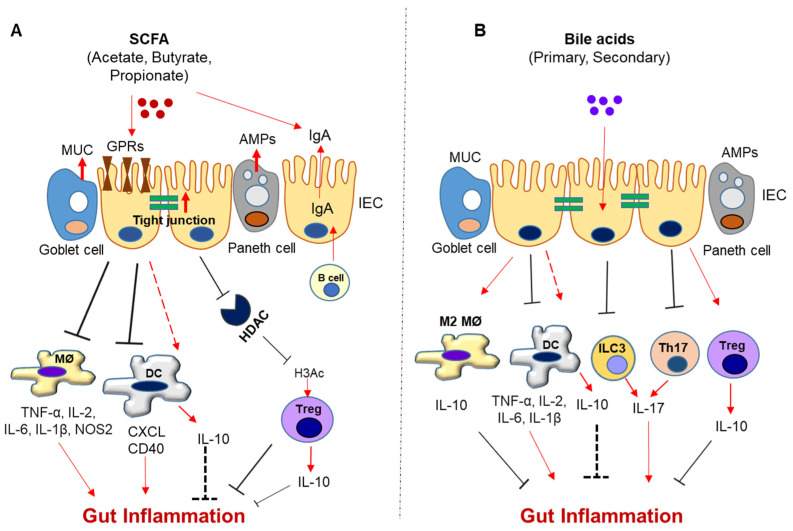
Possible mechanisms of short-chain fatty acids and bile acids positive effects on immune system in IBD. (**A**) Short-chain fatty acids (SCFAs) are fermented byproducts of dietary fiber. SCFAs are the ligands for G-protein receptors (GPRs) in which GPR activation upregulates mucin levels in goblet cells, antimicrobial peptides in Paneth cells, and tight junction proteins in intestinal epithelial cells. Moreover, SCFAs inhibit the secretion of pro-inflammatory cytokines (TNF-α, IL-2, IL-6, etc.) by macrophages, inhibit the expression of dendritic cell-migrated proteins (CXCL, CD40), and inhibit HDAC activity. HDAC inhibition allows for acetylation of histone 3 (H3Ac), which induces Treg differentiation and their secretion of anti-inflammatory cytokines, such as IL-10. Similarly, SCFA can promote DC-dependent anti-inflammatory IL-10 secretion. Finally, SCFAs induce IgA production from B cells. (**B**) Primary bile acids produced in the liver are metabolized by intestinal microbiota into secondary bile acids. Bile acids induce the polarization of macrophages and helper T cells into M2 macrophages and Treg, respectively, promoting anti-inflammatory IL-10 secretion. In addition, bile acids inhibit the secretion of pro-inflammatory cytokines (TNF-α, IL-2, IL-6, etc.) by DCs. Moreover, bile acids inhibit IL-17 secretion from ILC3 and Th17. Likewise, bile acids can promote DC-dependent IL-10 secretion. Overall, SCFA and bile acids reduce gut inflammation. SCFAs: Short-chain fatty acids, AMPs: Antimicrobial peptides, Mφ: Macrophages, DC: Dendritic cells, Tregs: T-Regulatory cells, Th1: T-helper 1, Th17: T helper 17, ILC3: Innate lymphoid cells type 3, IL: Interleukin, HDAC: Histone deacetylase, H3Ac: Acetylation of histone 3, TNF: Tumor necrosis factor, NOS2: Nitric oxide synthase 2, IgA: Immunoglobulin A, and CXCL: Chemokine (C-X-C motif) ligand. Red arrows denote activation, and black arrows denote inhibition.

**Figure 2 biomedicines-11-00294-f002:**
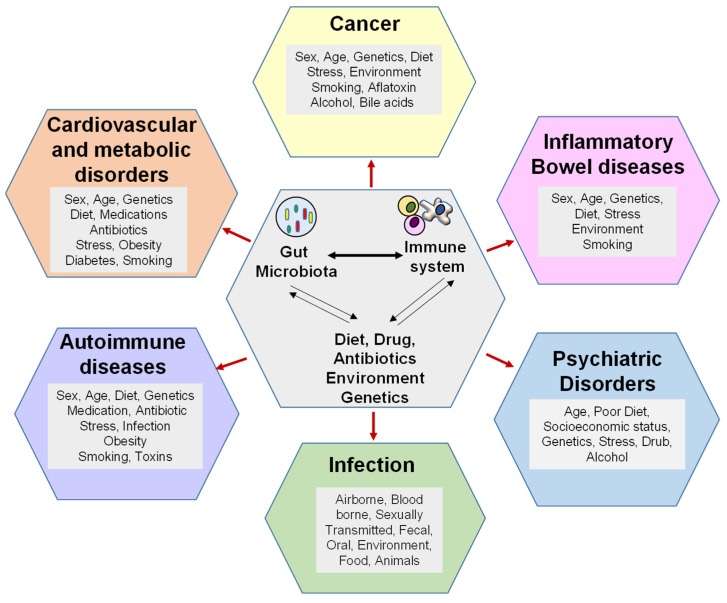
Gut microbiota dysbiosis gives rise to several pathophysiological conditions. Gut microbiota dysbiosis can be induced by diet, antibiotics, and genetic factors. Gut microbiota dysbiosis can cause and sustain cancers, such as colorectal cancer and hepatocellular carcinoma, along with inflammatory diseases, autoimmune conditions, and cardiometabolic disorders. Gut microbiota dysbiosis-induced immune dysregulation is another etiological factor for disease among the many others listed, including age, sex, and medication.

**Figure 3 biomedicines-11-00294-f003:**
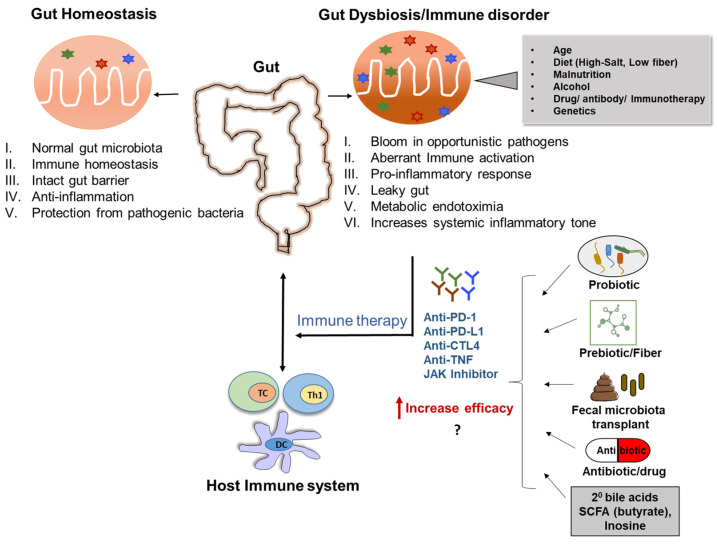
Modifying the abundance of gut microbiota population may influence the outcomes of immunotherapy. A healthy gut microbiome can increase the bioavailability and efficacy of drugs in the host. Dysbiosis, caused by several depicted factors, may decrease the efficacy of the therapeutic drugs, leading to poor therapeutic outcomes. Modifying gut microbiota could increase the effectiveness of certain immunotherapeutic drugs, such as anti-PD-1 antibody, anti-PD-L1 antibody, and anti-CTL4 antibody treatments. Gut microbiota can be changed by supplementation with either antibiotics, probiotics, prebiotics, secondary bile acids, short-chain fatty acids (e.g., butyrate), inosine, or fecal matter transplantation.

**Table 1 biomedicines-11-00294-t001:** Summary of gut microbiota–immune axis in various diseases.

Diseases	Reference	Findings
Gastrointestinal Infections	Singer et al., 2019 [222]	Provide resistance against colonization and invasion by pathobiont.
	Tovaglieri et al., 2019 [223]	Human gut microbiome metabolites induce expression of flagellin (a bacterial protein) increases EHEC motility and epithelial injury.
IBD	Lee and chang, 2021 [224]	Gut microbiota dysbiosis of IBD patients is consistently marked by an overgrowth in Proteobacteria.
	Furusawa et al., 2013 [110]	SCFA confers protection against IBD by maintaining gut barrier integrity, promoting Treg cell differentiation, and inhibiting histone deacetylases.
Colorectal carcinoma	Sepich-Poore et al., 2021 [225]	Generation of genotoxin such as Bacteroides fragilis toxin (Bft), cytolethal distending toxin (CDT), and colibactin.
	Hale et al., 2017 [226]	Bacterial-derived secondary bile acids and hydrogen sulfide promote proinflammatory milieu that increases CRC risk.
	Yeoh et al., 2020 [227]	Bacteria such as *F. nucleatum* can adhere to colon tumors and aggravate tumorigenesis.
Hepatocellular carcinoma	Lin et al., 1995 [228]	Systemic translocation of LPS promotes chronic liver injury and predisposes to HCC.
	Singh et al., 2018 [229]	Excess butyrate production promotes HCC progression.
	Yoshimoto et al., 2013 [230]	Secondary bile acids promote carcinogenesis and impede anti-tumor immunosurveillance in the liver.
Cardiometabolic disease	Cani et al., 2007 [231] Guasch-Ferré et al., 2017 [232] Millard et al., 2018 [233]	LPS and other microbial ligands drive low-grade chronic inflammation and predispose to CVD.
		Bacterial trimethylamine and its conversion to trimethylamine-N-oxide in the liver increases the risk of coronary artery disease, metabolic syndrome, stroke, and vascular inflammation.
Rheumatoid Arthritis	Scher et al., 2013 [234]	*Prevotella* spp. Abundance is positively associated with new-onset rheumatoid arthritis.
Allergic Diseases	Fazlollahi et al., 2018 [235] Bunyavanich et al., 2016 [236]	Gut microbiota dysbiosis increases risk for allergic disease, e.g., food allergy and asthma.

## Data Availability

Not applicable.

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
