# Peer review of "Crosstalk between Gut Microbiota and Host Immunity: Impact on Inflammation and Immunotherapy"

_biomedicines, 2023, doi:10.3390/biomedicines11020294_

Round 1

Reviewer 1 Report

This review article discusses the Crosstalk Between Gut Microbiota and Host Immunity: Impact on Inflammation and Immunotherapy. It is a well-organized and well-written review. However, please consider the following comments:

• Several reviews have been published related to the topic of this review (example: PMID: 32433595, PMID: 35469059 and others). It seems that there is some overlap in topics and conclusions between these reviews and the current article. Please highlight the novel aspects and discussions in this manuscript. The rationale, introduction and conclusion of the review could be modified to reflect this point.

·       The manuscript could benefit from a high-quality figure summarizing mechanistic pathways focusing on more recent data/ less reviewed disease areas.

·       Key novel findings are emerging with current research progresses to translation beyond the association between gut microbiota and disease. Consider highlighting these (example PMID: 36287379 and others) in this review and expand on the promises and challenges in immune-microbiome research.

·       It would seem more appropriate to write separate section of CVD, diabetes and hypertension instead of clumping them together as different diverse mechanisms underly them.

·       Related to the above comment, it would be helpful to expand on environmental factors including excess dietary salt and fiber intake.

·       There was some mention that microbiota impact efficacy of drugs but no references were included.

Author Response

Reviewer 1

This review article discusses the Crosstalk Between Gut Microbiota and Host Immunity: Impact on Inflammation and Immunotherapy. It is a well-organized and well-written review. However, please consider the following comments:

Response: Thank you for taking the time to review our manuscript. We appreciate your suggestions to improve our review and have addressed all your comments below.

  1. Several reviews have been published related to the topic of this review (example: PMID: 32433595, PMID: 35469059 and others). It seems that there is some overlap in topics and conclusions between these reviews and the current article. Please highlight the novel aspects and discussions in this manuscript. The rationale, introduction and conclusion of the review could be modified to reflect this point.

Response: Thank you for this important comment. We have highlighted the novel aspects in the rationale/abstract, introduction (see revised Section 1), and conclusion (see revised Section 10) sections of the review. One novel attribute that is weaved throughout the entire review is special emphasis on microbial-derived metabolites in host immunity (see New Sections 5.1 and 5.2) and pathophysiology (see revised Section 7). This includes detailed descriptions of new mechanisms to how the gut microbiota (and their metabolites) regulates and is regulated by host immunity (see revised Sections 2-4). Another novel aspect that makes our review unique is the in-depth discussion about gut microbiota influencing immunotherapy efficacy (see revised Section 8). Alongside, we delineate the current gut microbiota-targeted therapies in each mentioned disease (see revised Sections 6 and 7) and underscore the necessity to personalize these treatments to each pathology.

  1. The manuscript could benefit from a high-quality figure summarizing mechanistic pathways focusing on more recent data/ less reviewed disease areas.

Response: Thank you for this great suggestion. As this review has a focus on microbial-derived metabolites in host immunity, we designed a new figure (see New Figure 1A-B) that outlines the mechanisms to how short chain fatty acids and secondary bile acids modulate immune responses in the context of inflammatory bowel disease.  

  1. Key novel findings are emerging with current research progresses to translation beyond the association between gut microbiota and disease. Consider highlighting these (example PMID: 36287379 and others) in this review and expand on the promises and challenges in immune-microbiome research.

Response: Thank you for these comments. We fully agree that advances in gut microbiota research are highlighting more mechanistic and therapeutic findings that are beyond association results. Accordingly, we now detail the mechanistic and translational potential of gut microbiota-derived metabolites, probiotics, prebiotics, and fecal microbiota transplantation in each mentioned disease (see revised Section 7). Importantly, we underscore the necessity to personalize these treatments to each pathology because some interventions (e.g., butyrate supplementation) can alleviate some diseases while aggravate others. Furthermore, we have expanded on the promises/challenges in immune-microbiome research (see revised Section 9).

  1. It would seem more appropriate to write separate section of CVD, diabetes and hypertension instead of clumping them together as different diverse mechanisms underly them.

Response: Thank you for this suggestion. We have separated CVD (new Section 7.6), diabetes (new Section 7.7), and hypertension (new Section 7.8) into their own sections and expanded the discussion for each disease.

  1. Related to the above comment, it would be helpful to expand on environmental factors including excess dietary salt and fiber intake.

Response: Thank you for this great point. We have inputted new sections focused on how high-salt intake (new Subsection 6.3.1) and fiber intake (new Subsection 6.3.3) impacts the gut microbiota and their role in pathophysiology. Alongside, as per suggestion from Reviewer #3, we have included discussion on dietary polyphenols (new Subsection 6.3.2) to expand on environmental factors.

  1. There was some mention that microbiota impact efficacy of drugs but no references were included.

Response: Thank you for pointing this out to us. We apologize for the oversight. Please see our revised Section 8 on immunotherapy, which inputs the missing references and includes additional references as we have expanded our discussion to highlight the novel aspects for our review (see Response to Question 1).

Reviewer 2 Report

Dear Editor,

I have read with great interest the manuscript proposed by Campbell et al. entitled “Crosstalk Between Gut Microbiota and Host Immunity: Impact on Inflammation and Immunotherapy”. However, there are some issues that need to be addressed before further processing.

1.      The introduction does not cover crucial definitions of gut microbiota – I suggest citing these two recent papers (DOI-1: https://doi.org/10.1152/ajpgi.00161.2019 /DOI-2: https://doi.org/10.3390/jcm9113705).

2.      The sections 3 and 4, regarding the possible host-microbiota interactions are crucial for the manuscript and should be more extensive and have a summary image/table that summarize principal mechanisms.

3.      For the upcoming sections, I suggest dividing into macro-categories (e.g., cancer-related, autoimmune, infective, ecc), thus allowing better reading and understanding. I also suggest introducing a summary table of the findings. Also, for probiotics and FMT I suggest checking the section of DOI-1 to have an insight of real effects and possible side effects that should be reported in the manuscript.

Author Response

Reviewer 2

I have read with great interest the manuscript proposed by Campbell et al. entitled “Crosstalk Between Gut Microbiota and Host Immunity: Impact on Inflammation and Immunotherapy”. However, there are some issues that need to be addressed before further processing.

Response: Thank you for reviewing our manuscript and your prompt response. We appreciate your comments and suggestions that helped improve our manuscript.

  1. The introduction does not cover crucial definitions of gut microbiota – I suggest citing these two recent papers (DOI-1: https://doi.org/10.1152/ajpgi.00161.2019 /DOI-2: https://doi.org/10.3390/jcm9113705).

Response: Thank you for this suggestion. We have added appropriate definitions of the gut microbiota in the introduction (see revised Section 1), including citing the above-mentioned publications.

  1. 2.      The sections 3 and 4, regarding the possible host-microbiota interactions are crucial for the manuscript and should be more extensive and have a summary image/table that summarize principal mechanisms.

Response: Thank you for these insightful comments. We have expanded the concepts in revised Sections 3 and 4 with special emphasis on the mechanisms to how gut microbes and their metabolites regulate both innate and adaptive immunity. Moreover, we have updated our graphical abstract (see Page 1), which best summarizes sections 3 and 4. Also, as recommended by Reviewer #1, we have designed a New Figure 1A-B that depicts microbial metabolites modulation of host immunity, which highlights new findings mentioned in New Sections 5.1 and 5.2.

  1. For the upcoming sections, I suggest dividing into macro-categories (e.g., cancer-related, autoimmune, infective, ecc), thus allowing better reading and understanding. I also suggest introducing a summary table of the findings. Also, for probiotics and FMT I suggest checking the section of DOI-1 to have an insight of real effects and possible side effects that should be reported in the manuscript.

Response: These are excellent suggestions, thank you. We have categorized the relevant sections, including CVD, diabetes and hypertension as recommended by Reviewer #1. We also rearranged the ordering of the subsections in Section 7 (per suggested by Reviewer #3) so that the reading is more connected. Moreover, we have added the main effects and possible side effects of probiotics and FMT treatment in the new Subsection 6.3.3 and revised Section 6.2, respectively. Furthermore, we have generated a New Table 1 that summarizes the findings from revised Section 7.

Reviewer 3 Report

Campbel et al. review, in their article entitled Crosstalk between Gut Microbiota and Host Immunity, the past association of gut microbiota with various diseases and discuss inflammation caused by abnormalities in the gut microbiota and the impact of immunotherapy, which has recently been clinically applied to many diseases.

The article is generally concise, but there are some issues that require reorganization.

P.3 Unaspiringly, 

P.7 no surprise 

The above expressions are inappropriate for a review article and should be changed to "Reasonable" or similar.

P5 6.1

The example of antibiotics in the immediate postnatal period is given, but it would be better to give an example of how antibiotics alter the intestinal microbiota in adults, i.e., mature intestinal microbiota.

P6 6.2

FMT is an established treatment for CDI, but there have been cases of bloodstream infections derived from FMT (

NEJM) and that it is important to ensure safety.

P7 6.3

Regarding diet, it would be good to describe the positive effects of oriental diet or Chinese tea on intestinal microflora.

P9 7.5

If the conclusion is that changes in microbial metabolism in HCC are not consistent, more emphasis should be placed on other inflammatory bowel diseases, etc., beyond the detailed description in this part. Also, how about the relevance of psychiatric and other diseases?

P10 7.7

The part on gastrointestinal tract infections should be listed higher, e.g., after 7.2.

P11 9

Describe that 16S analysis is not sufficient and that more detailed analysis is currently underway, such as shotgun metagenomics and metabolomics.

Author Response

Reviewer 3

Campbell et al. review, in their article entitled Crosstalk between Gut Microbiota and Host Immunity, the past association of gut microbiota with various diseases and discuss inflammation caused by abnormalities in the gut microbiota and the impact of immunotherapy, which has recently been clinically applied to many diseases.

The article is generally concise, but there are some issues that require reorganization.

Response: Thank you for your critical review of our manuscript. Your suggestions and advice helped to improve our manuscript.

P.3 Unaspiringly, 

P.7 no surprise 

The above expressions are inappropriate for a review article and should be changed to "Reasonable" or similar.

Response: Thank you for bringing this to our attention. We have removed the words ‘unaspiringly’ and “no surprise’ from our manuscript and made the necessary changes.

P5 6.1

The example of antibiotics in the immediate postnatal period is given, but it would be better to give an example of how antibiotics alter the intestinal microbiota in adults, i.e., mature intestinal microbiota.

Response: Thank you for this comment. We have added examples of how antibiotics administered to adults alter their intestinal microbiota in the revised Section 6.1.

P6 6.2

FMT is an established treatment for CDI, but there have been cases of bloodstream infections derived from FMT (NEJM) and that it is important to ensure safety.

Response: Thank you for this insightful comment. We have mentioned this point of caution in our revised Section 6.2.

P7 6.3

Regarding diet, it would be good to describe the positive effects of oriental diet or Chinese tea on intestinal microflora.

Response: Thank you for this conceptual idea. We have added relevant information on Epigallocatechin-3-gallate (EGCG; a major catechin in green tea) to the New Subsection 6.3.2.

P9 7.5

If the conclusion is that changes in microbial metabolism in HCC are not consistent, more emphasis should be placed on other inflammatory bowel diseases, etc., beyond the detailed description in this part. Also, how about the relevance of psychiatric and other diseases?

Response: We appreciate and agree with your perspective on this matter. Accordingly, we have expanded information on IBD (see revised Section 7.3) and other pathologies mentioned in Section 7. Also, we have added a New Section 7.11 on psychiatric and neurodegenerative disorders in the revised manuscript.

P10 7.7

The part on gastrointestinal tract infections should be listed higher, e.g., after 7.2.

Response: Thank you for this comment. We moved gastrointestinal infections to be the revised Section 7.2 and we have rearranged other sections so that the reading is more connected. 

P11 9

Describe that 16S analysis is not sufficient and that more detailed analysis is currently underway, such as shotgun metagenomics and metabolomics.

Response: Thank you, we appreciate your suggestion. We have added the limitation of 16S analysis and describe the more relevant approaches for microbiota analysis in revised Section 9.

Round 2

Reviewer 1 Report

No additional comments

Reviewer 2 Report

The authors have addressed the reviewers comments. I believe the manuscript is now acceptable for publication.